# Dyslipidemia, Obesity, and Ethnicity in Mexican Children

**DOI:** 10.3390/ijerph182312659

**Published:** 2021-12-01

**Authors:** Paula Costa-Urrutia, Valentina Colistro, Valentina Franco-Trecu, Julio Granados, Rafael Álvarez Fariña, Martha Eunice Rodríguez-Arellano

**Affiliations:** 1Laboratorio de Medicina Genómica del Hospital Regional Lic. Adolfo López Mateos, ISSSTE, Ciudad de México 01030, Mexico; 2Departamento de Métodos Cuantitativos, Facultad de Medicina, Universidad de la República, Avda, General Flores 2125, Montevideo 11800, Uruguay; valentinacolistro@gmail.com; 3Departamento de Ecología y Evolución, Facultad de Ciencias, Universidad de la República, Iguá 4225, Montevideo 11400, Uruguay; vfranco-trecu@fcien.edu.uy; 4Departamento de Trasplantes, División de Inmunogenética, Instituto Nacional de Ciencias Médicas y Nutrición Salvador Zubirán, Avda, Vasco de Quiroga, Ciudad de México 14080, Mexico; julgrate@yahoo.com; 5Ciencias del Deporte, Universidad Anáhuac, Av. Universidad Anáhuac 46, Col. Lomas Anáhuac, Huixquilucan, Naucalpan de Juárez 52786, Mexico; planentrenar@gmail.com

**Keywords:** lipid, dyslipidemias obesity, ethnicity, children, Mexican

## Abstract

The aim of this study was to assess lipid disorders in children from five ethnic groups, both urban and indigenous, from northern and central Mexico. We measured the lipid profile to determine the ability of the body mass index (BMI) to discriminate an abnormally high lipid level using receiving operating characteristics (ROC). We analyzed the association and interaction of obesity and ethnicity with lipid disorders using generalized linear models in 977 children. The highest prevalence of lipid disorders (high TG, high TC, high LDL, high APOB, and dyslipidemia) was found in central Mexico-Mexico City and urban northern Mexico. The BMI performed better at predicting low HDL in Seris, a northern indigenous group (0.95, CI: 0.69–0.85), and Mexico City (0.75, CI: 0.69–0.82), and high LDL in Puebla (central Mexico, 0.80, CI: 0.69–0.85). Obesity significantly (*p* < 0.05) increases lipid disorders by around two times (OR~2) for almost all lipid markers. Obesity and ethnic interaction increase the lipid disorders by more than five times for different lipid markers and ethnic groups (high total cholesterol OR = 5.31; low HDL OR = 5.11, and dyslipidemia OR = 5.68). Lipid disorders are not restricted to children with high BMIs, but obesity exacerbates these. The emerging lipid disorder risk depends on the ethnic group.

## 1. Introduction

Lipid disorders and obesity in childhood are associated with an earlier onset and greater risk of many chronic disorders in adults such as cardiovascular disease (CVD), metabolic syndrome, type 2 diabetes [1,2,3], and premature death [4]. Thus, early screening of abnormal lipids levels is crucial for the prevention and intervention of chronic diseases [5].

Mexico ranks as one of the top countries worldwide for childhood obesity, with a mean national prevalence of 34.9% for schoolchildren [6]. Among adults, CVD has been a significant cause of death since 1990, with a mortality rate of 196 deaths per 100,000 that year. The Global Burden of Disease study estimated that from 1990 to 2017, CVD accounted for 22.7% of all deaths in Mexico [7]. The leading CVD risk factors are high systolic blood pressure, dietary risks, high levels of low density lipoprotein (LDL), cholesterol, a high body mass index (BMI), and a high fasting plasma glucose level [8]. Both trends—adult CVD and childhood obesity—make children especially vulnerable to the development of chronic disease and premature death.

Genetic, cultural, and lifestyle differences among ethnicities, at least, factor into differences in obesity prevalence and other metabolic diseases in Mexico [9,10,11]. However, we scarcely know the prevalence of lipid disorders among children or how they may differ among ethnicities.

Although lipid disorders are directly determined by the lipid profile, non-invasive and easy anthropometric measurements are feasible alternatives for assessing abnormal lipid levels in early stages because of the direct relationship with childhood obesity [1,12,13]. This is particularly true for remote indigenous groups, where there is greater vulnerability due to a lack of medical services and clinical laboratories [14].

Thus, the aims of this study were to: (a) estimate the prevalence of several lipids’ markers in children from five ethnic groups (three urban and two indigenous) from northern and central Mexico, (b) assess the ability of the BMI and body fat percentage (BFP) to discriminate lipid disorders, and (c) analyze the association of obesity and ethnicity with lipid disorders.

## 2. Materials and Methods

This study included Mexican children, aged from 5 to 13 years old, from northern (Sonora State) and central Mexico (Puebla State and Mexico City). Children from central Mexico-Mexico City (CM-Mexico City) came from of the Childhood Obesity Cohort of the Healthy Childhood Project, carried out at the Genomic Medicine Laboratory at Hospital Regional Lic. Adolfo López Mateos, Instituto de Seguridad y Servicios Sociales de los Trabajadores del Estado, ISSSTE (Institute of Security and Social Services of State Workers) [15]. Children from northern Mexico (NM) and central Mexico-Puebla (CM-Puebla) were part of the *Por tu Salud* Project, aiming to elucidate the genetic and environmental risk factors of childhood obesity. All children from the *Por tu Salud* Project attended seven schools, four in Sonora, Hermosillo (the capital of the state; NM-Urban), one in CM-Puebla, and two indigenous schools from Sonora: Seris (NM-Seris) and Yaquis (NM-Yaquis). Indigenous schools are characterized by bilingualism and biculturalism, where at least one indigenous language and culture of a particular Amerindian group is taught [10]. The Secretaría de Educación de Cultura del Estado Sonora (Sonora State Secretary of Education and Culture) and Secretaría de Educación Pública del Estado Puebla (Puebla State Public Secretary of Education) oversaw the participant schools. All children voluntarily participated, and their parents authorized their participation by signing informed consent. The projects were approved by the Research and Ethics Committee of Regional Hospital Lic. Adolfo López Mateos, ISSSTE under the registry numbers: 433.2016, 315.2018, and 447.2016 for the *Por tu Salud* Project in Sonora, Puebla, and COIPIS, respectively.

### 2.1. Anthropometric and Lipid Measurements

The weight and height were measured by trained technicians. The children were barefoot and wore light clothes. The accuracy of the stadiometer was ±0.1 cm and ±0.01 kg. The BMI was calculated (kg/m^2^), as well as the BMI z-score (BMIz). Only children from Mexico City were also measured by electrical bioimpedance to estimate the BFP, using InBody J10 tetrapolar equipment, of three frequencies (5, 50, and 250 kHz) and anthropometry [16].

Blood samples were taken in the morning, after a 10 h fast, to measure the serum lipid profile. Measures registered: triglycerides (TG) measured in mg/dL, total cholesterol (TC) in mg/dL, low density lipoprotein cholesterol (LDL) in mg/dL, high density lipoprotein cholesterol (HDL) in mg/dL, apolipoprotein A1 (ApoA1) in mg/dL, and apolipoprotein B (ApoB) in mg/dL. The samples were kept in coolers for no longer than two hours before further processing and were analyzed by Miura Co., Ltd., Matsuyama, Japan.

### 2.2. Classification of Obesity by BMI and BFP

We classified obesity (OB) following the World Health Organization cut-off value for BMI in relation to sex and age [17,18]. For the BFP, we chose the health-related definition of OB—BFP ≥30 for girls and BFP ≥25 for boys—as these cut-offs have been significantly associated with cardiovascular risk factors in children and adolescents and were previously used in this same cohort [15,19].

### 2.3. Classification of Lipid Disorders

We classified the lipid levels according the borderline-high criteria of the Expert Panel on Integrated Guidelines for Cardiovascular Health and Risk Reduction on Children and Adolescents of the National Heart, Lung and Blood Institute (NHLBI) [5,20] for increased cardiovascular risk as follows: high TG: ≥75 mg/dL (0–9 years old) and ≥90 mg/dL (10–19 years old); high TC: ≥170 mg/dL; low HDL: ≤45 mg/dL; high LDL: ≥110 mg/dL; low ApoA1: ≤120 mg/dL; and high ApoB: ≥90 mg/dL. As we had no available information on ApoA1 and ApoB for CM-Puebla, dyslipidemia was defined as the presence of one or more of the following: abnormal TG, TC, HDL, or LDL levels.

### 2.4. Data Analysis

We present descriptive results for all variables analyzed (TG, TC, HDL, LDL, AopA1, ApoB, BMI, BMIz, and BFP) as the median and interquartile ranges stratified by the gender and ethnicity because the variables did not show a normal distribution after running the Shapiro–Wilk test. We performed gender pairwise comparisons using the Mann–Whitney U test in XLSTAT software (Data Analysis and Statistical Solution for Microsoft Excel, Addinsoft, Paris, France, 2017). The lipid disorders and dyslipidemia prevalence were presented for the total sample by ethnic group and OB.

The discriminating ability of BMIz and BFP to detect lipid disorders and dyslipidemia was tested by estimating the area under the receiver operating characteristic (ROC) curves (AUC) using XLSTAT software. The AUC for the ROC curves represents the proportion of true positive cases (sensitivity) as a function of the proportion of false positives cases (corresponding to 1-specificity) [21]. The AUC ranges from 0 to 1, where 1 is a perfect score and 0.5 indicates results no better than chance. This reflects whether a randomly selected participant with lipid disorders has higher BMI and BFP values than those of a randomly selected participant. The AUC of the ROC curves was assessed by stratifying by ethnic group.

We used a generalized linear model (GLM) with logistic function to evaluate the influence of OB classified by BMI on each lipid marker as a dependent variable. The explanatory variables included in the models were age, gender, OB, and ethnicity, as well as the OB*Gender and OB*Ethnicity interactions. In addition, we used GLM logistics to evaluate the influence of OB classified by BFP on each lipid marker as a dependent variable for CM-Mexico City. The explanatory variables included in the models were age, gender, OB, and OB and Gender interaction. For all models with OB classified by both BMI and BFP, the initial models contained all single effects and interactions. Model simplification was done by stepwise deletion of the least significant terms. We evaluated the relative performance of models using the Akaike information criteria (AIC) [22] and selected the model with the lowest AIC. Odds ratios (ORs), confidence intervals at 95% (CI 95%), and *p*-values were estimated for all logistic models. All GLMs were carried out using the R programming language and free software environment for statistical computing and graphics [23].

## 3. Results

### 3.1. Population Study and Lipid Disorder Prevalence

This study included a total of 977 Mexican children (572 boys and 405 girls), aged from 5 to 13 years old, from CM-Mexico City (*n* = 325), CM-Puebla (*n* = 302), NM-Urban (*n* = 148), and two indigenous communities: NM-Yaquis (*n* = 135) and NM-Seris (*n* = 67). The descriptions of anthropometric variables and cardiometabolic markers are presented in Table 1. The overall median age was nine years old for both boys and girls. Pairwise comparisons for anthropometric and lipid disorders by ethnicity showed few significative differences based on gender (Table 1). Thus, the rest of the analyses were not stratified by gender.

The highest prevalence of OB, high TG, high TC, high LDL, high APOB, and dyslipidemia were found in CM-Mexico City and NM-Urban, while the highest prevalence of low HDL and low ApoA1 were found in CM-Puebla and NM-Yaquis, respectively (Table 2).

### 3.2. The Ability of BMI and BFP to Discriminate Lipid Disorders

For the overall sample, the AUC-ROCs of BMI for each lipid disorder ranged from 0.50 to 0.68. By ethnic group, the AUC-ROCs ranged from 0.50 to 0.95. Among them, the AUC-ROCs for low HDL in CM-Mexico City, high LDL in CM-Puebla, and low HDL, high TG, and dyslipidemia in Seris were greater than or equal to 0.75 (Table 3).

The AUC-ROCs of BFP performed for lipid disorders in CM-Mexico City alone, ranged from 0.51 to 0.71. Among them, the AUC-ROCs for low HDL and high TG were higher than or equal to 0.70. In general, the AUC-ROCs for BMI and BFP performed similarly. Comparing the AUC-ROCs between BFP and BMI showed that BFP generally performed slightly better, but significantly better for high TC, high TG, and high ApoB when compared to BMI. On the contrary, BMI performed slightly better for low HDL when compared to BFP (Appendix A).

### 3.3. Obesity’s Association with Lipid Disorders

The GLM showed a significant association between OB classified by BMI with each of the following lipid markers: high TG, low HDL, low ApoA, high ApoB, and dyslipidemia. Every model showed OR > 2.4, except for ApoB, in which the OR was closer to 2 (OR = 1.97). Both CM-Mexico City and NM-Urban showed a higher risk of high TG, high TC, and dyslipidemia, while CM-Puebla showed a higher risk of low HDL and NM-Yaquis showed a higher risk of low ApoA1 compared to the other four ethnicities, respectively (Table 4, Figure 1). As shown in Figure 1, the OB (shown in blue) increased the probability of lipid disorder (except for high LDL) with respect to normal weight (shown in red).

The best models for high TC, low HDL, and dyslipidemia kept the OB*Ethnicity interaction. The OB*Ethnicity showed that children with OB from NM-Yaquis, NM-Seris, and CM-Puebla had more than five times (OR > 5.0) the risk of high TC, low HDL, and dyslipidemia (Table 4).

OB classified by PBF, when measured in CM-Mexico City, showed a significant association with all the lipid markers. The associations were higher for high TG (OR = 3.78, *p* < 0.0001), low HDL (OR = 3.87, *p* = 0.0001), and dyslipidemia (OR = 4.21, *p* < 0.0001; Appendix A).

## 4. Discussion

This study estimated the prevalence of several lipid disorders in 977 schoolchildren from five groups from central and northern Mexico. We assessed the ability of the BMI and BFP to discriminate lipid disorders and dyslipidemia and we analyzed the interaction between lipid disorders and OB. We found higher lipid disorders in children from CM-Mexico City and NM-Urban compared to CM-Puebla, NM-Yaquis, and NM-Seris. However, the performance of AUC-ROC and the lipid disorder risk varied according to the ethnic group.

More than 50% of the children from the five ethnic groups (except Seris, with 46.3%) had dyslipidemia. The highest prevalence of dyslipidemia was in CM-Mexico City (75.5%) and NM-Urban (76.4%), in which three-quarters of each group had at least one borderline-high lipid level. Among each marker, the highest prevalence of high TG and high TC was also found in CM-Mexico City (high TG = 60.9%, high TC = 54.2%) and NM-Urban (high TG = 58.1%, high TC = 51.4%) (Table 2). Our results in the child population are in line with the adult population’s trend. Mexico City and Sonora in northern Mexico are two states with the highest prevalence of cardiovascular disease [8]. We found the greatest low-HDL prevalence in CM-Puebla (27.5%), which is also in line with the adult population’s trend. In Puebla, hypercholesterolemia is the second cardiovascular risk factor, after being overweight or obese [24].

The biggest cities, like Mexico City and the states in northern Mexico, have undergone the typical food transformation of most industrialized countries. Among the urban communities and indigenous groups, there are different levels of westernized lifestyles [25,26]. In particular, the Yaqui group currently represents a population of 45,000 individuals divided into eight villages, who live in rural and urban communities, showing a high genetic and cultural mixture [10,27,28]. Currently, adult Yaquis are the most likely group to be overweight or obese among the populations in northern areas (87.6%) [29]. Although NM-Yaqui children showed lower OB prevalence than NM-urban children, NM-Yaquis presented the highest prevalence of low ApoA1 (17.8%), one of the best predictors of myocardial infarction [22] and a clear sign of cardiovascular risk (Table 2).

Our results showed that the ability of BMI to predict lipid disorders performed differently according to the ethnic group. The BMI performed better at predicting low HDL in CM-Mexico City (0.75, CI: 0.69–0.82) and NM-Seris (0.95, CI: 0.90–1.00), and high LDL (0.80, CI: 0.69–0.85) in Puebla. Our results are in line with a previous study carried out in children from CM-Mexico City (*n* = 142), where AUC-ROCs for BMI were also better for low HDL (0.70, CI: 0.57–0.83) and high TG (0.73, CI: 0.64–0.82) compared to high LDL (0.68, CI: 0.55–0.82) and high TC (0.65, CI: 0.50–0.81) [30]. In addition, our results also showed that BFP performed similarly to BMI in CM-Mexico.

Among ethnicities, BMI performed better at predicting most of the lipid markers (low HDL 0.95, CI: 0.90–1.00; high TG 0.82, CI: 0.73–0.93; high ApoB 0.77, CI: 0.61–0.94; dyslipidemia 0.76, CI: 0.64–0.97) in NM-Seris. There is no straightforward explanation for why the AUC-ROCs outperform in Seris compared to the rest of the groups, but probably, the answer lies in a mixture of genetics, dietary patterns, and lifestyle. The Seri group is an isolated hunter-gatherer and fishing community in the desert of northwestern Mexico [31], with different Amerindian ancestry from the rest of the groups [10,11]. Seafood is an important part of the Seri diet [32]. Seafood has well-known nutritional and health attributes because it is rich in polyunsaturated fatty acids, minerals, and vitamins, and has low saturated fat [33]. In addition, the Seris community (Punta Chueca) is a small village in which the children are free to move about on foot; they walk to school and play in the street. The potentially higher physical activity and healthy food intake of Seris children lead us to suggest that lipid disorders are more directly associated with high BMI, and thus, BMI becomes a better predictor compared to the rest of the groups.

Apart from the mentioned findings, the rest of the results of the AUC-ROCs ranged from 0.50 to 0.68 and were in line with most studies, finding that anthropometric indices have a poor ability to discriminate abnormal lipids levels. In a recent study in children from China (*n* = 65,347), the AUC-ROCs for BMI poorly predicted high TC (0.55, CI: 0.53–0.57), high LDL (0.60, CI: 0.57–0.62), low HDL (0.59, CI: 0.58–0.61), and high TG (0.66, CI: 0.65–0.68) [33]. Furthermore, the last extensive meta-analysis in children and adolescents (*n* = 169,630) from the American continent, Europe, Asia, and Africa supports this tendency. AUC-ROCs for BMI and other anthropometric parameters (waist circumference and waist-height ratio) ranged from 0.50 to 0.74 for high TC, high TG, low HDL, and high LDL [34].

Our result suggests that most lipid disorders are not restricted to children with high BMIs. This tendency is alarming but not surprising. In Guerrero, a central Mexican state, 44.3% (total *n* = 225) of schoolchildren with OB presented metabolic syndrome, while one-third of children with a normal weight showed at least one of the more frequent components of the metabolic syndrome: high TG or low HDL [35]. Although lipid disorders are not restricted to children with high BMIs, the OB condition exacerbates it. Children with OB are at almost two times the risk of high ApoB (OR = 1.97), and around two and a half times the risk for high TG (OR = 2.45), low ApoA1 (OR = 2.57), and dyslipidemia (OR = 2.54), and more than four and half times the risk for low HDL (OR = 4.75). The aforementioned general tendencies increase according to OB and ethnicity interactions. NM-Yaqui, NM-Seris, and CM-Puebla children have more than five times the risk of high TC (OR = 5.31), low HDL (OR = 5.11), and dyslipidemia (OR = 5.68), respectively, with the presence of OB (Table 4). This suggests that the transition to Westernization has caused the typical health problems of the Western world, but in Mexico, lipid disorders differ in their emergence among ethnic groups [31,36,37].

This study has limitations that are worthy of mention. The numbers of NM-Urban, NM-Seris, and NM-Yaqui children were considerably lower than those of the central Mexicans. Furthermore, this study did not collect information on potential confounders such as dietary patterns, lifestyle, or family history of hypercholesterolemia. As such, this only allowed us to generate a hypothesis about the causes of the emergence of different lipid disorders. However, suggestions were also made based on direct observation after several months of working in the communities.

## 5. Conclusions

This study reported the prevalence of lipid disorders of the most standard lipid markers in clinical practice, the predictability under high BMI and PBF, and the association with OB and ethnicity interaction. More than 50% of the children from the five ethnic groups, except Seris, with 46.3%, had at least one lipid disorder, which is a red flag from the public health framework. BMI could be used as a good predictor for health professionals of high TG and low HDL in CM-Mexico and NM-Seris and high LDL in CM-Puebla. BMI and BFP performed similarly in CM-Mexico City. Obesity increases the risk by around two times of all lipid disorders (except LDL) and more than four and a half times for low HDL. The emerging lipid disorders depend on the ethnic group, suggesting that current and future health problems could have different triggers depending on the diversity of genetics, environment, dietary patterns, and lifestyle. Our results are relevant to help policymakers and health professionals to make shifts toward community intervention work that could be promising for children’s health.

## Figures and Tables

**Figure 1 ijerph-18-12659-f001:**
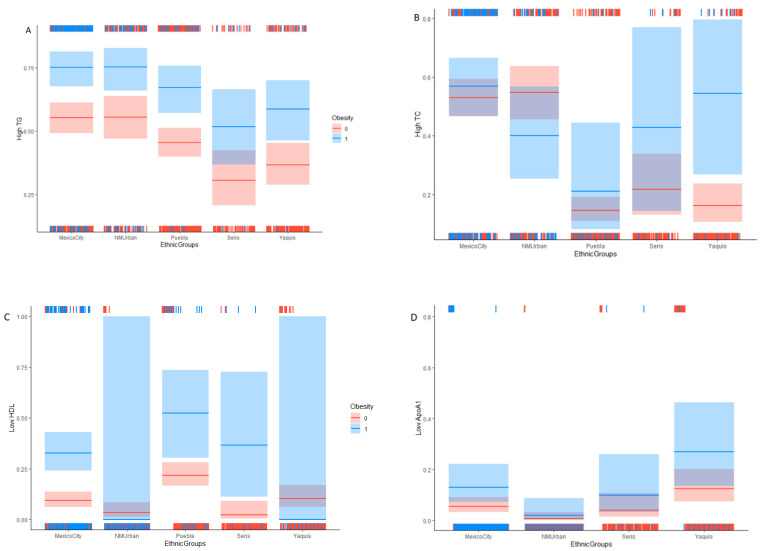
Effect of obesity and the ethnic groups on the probability of lipid disorders in northern and central Mexican child Obesity, 1 = obesity in blue, 0 = normal weight in red (**A**) high TG (Triglycerides), (**B**) high TC (total cholesterol), (**C**) low HDL (high density lipoprotein cholesterol), (**D**) low ApoA1 (apolipoprotein A1), (**E**) high ApoB (apolipoprotein B), (**F**) dyslipidemia. All lipids were measured in mg/dL.

**Table 1 ijerph-18-12659-t001:** Demographic, anthropometric, and lipid profiles of schoolchildren from northern (NM) and central Mexico (CM) for the overall sample and by ethnic group.

	Boys, *n* = 572		Girls, *n* = 405		
All, *n* = 977	Median	IQR	Median	IQR	*p*
Age	9.0	4.0	9.0	3.0	0.29
BMI	17.6	6.7	17.0	4.6	0.01
BMIz	−0.3	1.4	−0.2	1.1	0.60
TG	87.0	61.9	85.5	51.0	0.91
TC	157.8	44.5	155.0	38.0	0.10
HDL	56.5	15.4	55.1	17.3	0.27
LDL	83.5	30.6	81.1	26.0	0.20
ApoA1	153.0	52.0	153.0	58.0	0.23
ApoB	86.0	34.6	87.5	27.8	0.38
CM-Mexico City, *n* = 325					
Age	9.0	4.0	9.0	4.0	0.42
BMI	20.4	8.3	18.5	5.4	0.02
BMIz	0.3	1.7	0.3	1.3	0.75
TG	102.0	75.0	102.0	52.5	0.71
TC	173.0	46.0	169.5	35.5	0.05
HDL	56.1	16.1	57.2	20.2	0.91
LDL	92.8	37.9	89.6	28.1	0.09
ApoA1	152.0	42.0	142.0	34.3	0.01
ApoB	82.0	29.0	82.0	25.5	0.94
CM-Puebla, *n* = 302					
Age	9.0	3.0	9.0	3.0	0.52
BMI	16.6	3.4	16.7	3.5	0.86
BMIz	0.5	0.8	0.3	0.8	0.02
TG	77.1	47.7	81.8	46.5	0.19
TC	145.3	31.4	141.5	32.5	0.82
HDL	54.8	15.4	49.7	14.3	0.01
LDL	79.8	19.2	79.5	27.0	0.21
NM-Urban, *n* = 148					
Age	8.0	3.0	9.0	3.0	0.47
BMI	18.3	6.1	17.9	6.6	0.48
BMIz	01.0	1.3	01.0	1.4	0.47
TG	89.0	59.5	89.5	71.0	0.80
TC	167.0	48.0	170.5	37.0	0.61
HDL	63.6	16.6	67.4	16.3	0.38
LDL	84.0	33.3	85.4	25.1	0.97
ApoA1	194.0	31.5	191.0	44.3	0.59
ApoB	100.3	21.9	94.0	21.8	0.15
NM-Yaquis, *n* = 135					
Age	9.0	4.0	9.0	5.0	0.57
BMI	16.2	4.1	16.2	4.0	0.86
BMIz	−0.6	0.9	−0.4	0.9	0.50
TG	79.0	51.5	88.0	38.0	0.13
TC	145.5	32.0	151.0	28.0	0.16
HDL	55.7	11.9	52.7	11.1	0.24
LDL	72.4	17.5	77.1	18.1	0.06
ApoA1	139.5	29.3	135.0	22.0	0.14
ApoB	68.5	21.4	74.0	14.0	0.07
NM-Seris, *n* = 67					
Age	9.5	3.0	10.0	2.3	0.72
BMI	17.6	4.0	17.3	4.0	0.89
BMIz	−0.5	0.8	−0.4	1.0	0.75
TG	77.0	44.0	73.0	21.8	0.71
TC	152.0	36.3	148.0	29.3	0.56
HDL	54.5	6.3	54.4	9.0	0.81
LDL	68.3	21.7	72.3	15.6	0.32
ApoA1	145.0	19.0	139.0	28.3	0.35
ApoB	65.0	24.6	67.5	14.8	0.40

Interquartile range (IQR) and *p*-value (*p*) from Mann-Whitney U test gender pairwise comparisons. Body mass index (BMI), BMI z-score (BMIz), triglycerides (TG, mg/dL), total cholesterol (TC, mg/dL), high density lipoprotein (HDL in mg/dL), low density lipoprotein (LDL in mg/dL), apolipoprotein A1, (ApoaA, mg/dL), apolipoprotein B, (ApoB mg/dL).

**Table 2 ijerph-18-12659-t002:** Prevalence of lipid disorders among schoolchildren in central and northern Mexico for the overall sample (All) by ethnic group and by obesity (OB).

Prevalence	All *n* = 977	CM-Mexico City *n* = 325	CM-Puebla *n* = 302	NM-Urban *n* = 148	NM-Yaquis *n* = 135	NM-Seris *n* = 67
OB %	17	28.6	6.3	23.6	8.14	20.4
High TG, *n* (%)	496 (50.7)	198 (60.9)	138 (45.7)	86 (58.1)	51 (37.8)	23 (34.3)
% High TG with OB	23.0	34.8	10.9	24.4	17.4	21.7
High TC, *n* (%)	339 (34.7)	176 (54.2)	45 (14.9)	76 (51.4)	26 (19.3)	16 (23.9)
% High TC with OB	23.6	30.1	8.9	18.4	23.1	18.8
Low HDL, *n* (%)	166 (17.0)	58 (17.8)	83 (27.5)	5 (3.4)	15 (11.1)	5 (7.5)
% Low HDL with OB	28.3	56.9	13.3	0.0	0	60.0
High LDL *n* (%)	132 (13.5)	84 (25.4)	22 (7.3)	20 (13.5)	4 (3.0)	2 (3.0)
% High LDL with OB	28.0	33.3	18.2	25.0	0	0
Low ApoA1, *n* (%)	58 (8.6)	27 (8.3)	na	2 (1.4)	24 (17.8)	5 (7.4)
% Low ApoA1 with OB	20.7	33.3	na	0.0	4.1	40.0
High ApoB, *n* (%)	242 (35.8)	118 (36.3)	na	93 (62.8)	22 (16.3)	9 (13.4)
% High ApoB with OB	31.0	37.3	na	24.7	27.3	22.2
Dyslipidemia, *n* (%)	644 (66.0)	255 (75.5)	175 (58.0)	113 (76.4)	70 (52.0)	31 (46.3)
% Dyslipidemia with OB	21.7	32.2	10.3	22.1	12.9	19.4

Lipid disorders (high or low levels) were classified according to the borderline-high criteria of the Expert Panel on Integrated Guidelines for Cardiovascular Health and Risk Reduction in Children and Adolescents of the National Heart, Lung and Blood Institute. na, information for ApoaA and ApoB in CM-Puebla is not available.

**Table 3 ijerph-18-12659-t003:** The ability of body mass index to discriminate lipid disorders. Areas under the ROC curve (AUCs), 95% confidence intervals (CI), and *p*-values (*p*) of the body mass index for cardiometabolic risk in Mexican children, for the overall (All) sample and by ethnic group.

AUC (95%CI), *p*						
	All	CM-Mexico City	CM-Puebla	NM-Urban	NM-Yaquis	NM-Seris
High TG	0.68 (0.65–0.71), <0.001	0.68 (0.62–0.74), <0.001	0.67 (0.60–0.73), <0.001	0.55 (0.45–0.64), 0.322	0.66 (0.56–0.76), 0.002	0.82 (0.73–0.93), <0.001
High TC	0.60 (0.56–0.66), <0.001	0.54 (0.48–0.61), 0.183	0.57 (0.47–0.66), 0.144	0.47 (0.37–0.56), 0.464	0.53 (0.39–0.67), 0.652	0.62 (0.46–0.79), 0.146
Low HDL	0.64 (0.59–0.69), <0.001	0.75 (0.69–0.82), <0.001	0.66 (0.60–0.73), <0.001	0.46 (0.28–0.65), 0.700	0.50 (0.26–0.60), 0.411	0.95 (0.90–1.00), <0.001
High LDL	0.61 (0.56–0.69), <0.001	0.59 (0.52–0.66), 0.010	0.80 (0.69–0.85), <0.001	0.50 (0.37–0.64), 0.964	0.50 (0.16–0.64), 0.412	0.67 (0.30–1.00), 0.352
Low ApoA1	0.50 (0.40–0.56), 0.562	0.56 (0.42–0.67), 0.233	na	0.53 (0.32–0.73), 0.782	0.50 (0.33–0.58), 0.585	0.74 (0.47–1.00), 0.081
High ApoB	0.64 (0.60–0.68), <0.001	0.60 (0.53–0.62), 0.003	na	0.60 (0.50–0.69), 0.063	0.68 (0.54–0.81), 0.012	0.77 (0.61–0.94), 0.001
Dyslipidemia	0.68 (0.64–0.71), <0.001	0.68 (0.61–0.76), <0.001	0.64 (0.58–0.70), <0.001	0.53 (0.41–0.65), 0.141	0.57 (0.48–0.67), 0.146	0.76 (0.64–0.87), <0.001

Lipid disorders (high or low levels) were classified were according to the borderline-high criteria of the Expert Panel on Integrated Guidelines for Cardiovascular Health and Risk Reduction in Children and Adolescents of the National Heart, Lung and Blood Institute. na, information for ApoA1 and ApoB in CM-Puebla is not available.

**Table 4 ijerph-18-12659-t004:** Lipid disorder association analysis with obesity (OB) of central and northern Mexican schoolchildren from five ethnic groups, providing the estimate, standard error (SE), Z-statistic (Z) odds ratio, 95% confidence interval (CI 95%), and *p*-value (*p*).

High TG	Estimate	SE	Z	OR (CI 95%)	*p*
Intercept	−0.95	0.32	−2.99	0.39 (0.21, 0.72)	0.003
Age	0.13	0.03	3.93	1.14 (1.07, 1.21)	˂0.001
OB	0.9	0.19	4.61	2.45 (1.68, 3.62)	˂0.001
NM-Urban	0.01	0.21	0.05	1.01 (0.67, 1.52)	0.960
CM-Puebla	−0.39	0.17	−2.33	0.67 (0.48, 0.94)	0.020
NM-Seris	−1.04	0.29	−3.6	0.36 (0.20, 0.62)	˂0.001
NM-Yaquis	−0.76	0.22	−3.48	0.47 (0.32, 0.72)	˂0.001
High TC					
Intercept	0.12	0.13	0.92	1.13 (0.87, 1.46)	0.360
OB	0.16	0.25	0.65	1.17 (0.72, 1.91)	0.520
NM	0.07	0.23	0.32	1.08 (0.69, 1.70)	0.750
CM-Puebla	−1.90	0.21	−8.86	0.15 (0.10, 0.23)	<0.001
NM-Seris	−1.41	0.34	−4.14	0.25 (0.12, 0.46)	<0.001
NM-Yaquis	−1.77	0.28	−6.38	0.17 (0.11, 0.29)	<0.001
NM*OB	−0.76	0.46	−1.64	0.47 (0.19, 1.15)	0.100
CM-Puebla*OB	0.29	0.64	0.46	1.34 (0.34, 4.36)	0.650
NM-Seris*OB	0.84	0.86	0.97	2.31 (0.39, 12.64)	0.330
NM-Yaquis*OB	1.67	0.70	2.39	5.31 (1.35, 21.81)	0.021
Low HDL					
Intercept	−3.08	0.49	−6.31	0.05 (0.02, 0.12)	<0.001
Age	0.09	0.05	1.94	1.09 (1.00, 1.20)	0.051
OB	1.56	0.31	5.09	4.74 (2.62, 8.72)	<0.001
Gender (girls)	0.47	0.19	2.49	1.60 (1.11, 2.31)	0.011
NM-Urban	−1.03	0.51	−2.02	0.36 (0.12, 0.92)	0.043
CM-Puebla	1.00	0.26	3.88	2.72 (1.66, 4.57)	<0.001
NM-Seris	−1.46	0.75	−1.94	0.23 (0.04, 0.82)	0.052
NM-Yaquis	0.11	0.35	0.31	1.11 (0.55, 2.19)	0.763
NM-Urban*OB	−16.07	64.21	−0.02	0.01 (0.01, 61.55)	0.982
CM-Puebla*OB	−0.18	0.58	−0.32	0.83 (0.27, 2.64)	0.752
NM-Seris*OB	1.63	1.10	1.48	5.11 (0.61, 15.37)	0.143
NM-Yaquis*OB	−17.40	80.06	−0.02	0.02 (0.01, 6.77)	0.999
High LDL					
Intercept	−1.78	0.46	−3.84	0.17 (0.07, 0.41)	˂0.001
Age	0.09	0.05	1.91	1.10 (1.10, 1.20)	0.062
Gender (Girls)	−0.45	0.22	−2.05	0.64 (0.41, 0.97)	0.042
NM	−0.61	0.28	−2.17	0.54 (0.31, 0.93)	0.033
CM-Puebla	−1.37	0.26	−5.31	0.25 (0.15, 0.41)	˂0.001
Seris	−2.33	0.73	−3.18	0.10 (0.02, 0.32)	0.001
Yaquis	−2.35	0.52	−4.48	0.10 (0.03, 0.24)	˂0.001
Low ApoA1					
Intercept	−2.84	0.28	−10.02	0.06 (0.03, 0.1)	˂0.001
OB	0.94	0.42	2.26	2.57 (1.10, 5.77)	0.022
Gender (Girls)	0.92	0.33	2.82	2.51 (1.33, 4.8)	˂0.001
NM	−1.98	0.75	−2.64	0.14 (0.02, 0.48)	0.011
Seris	−0.3	0.53	−0.57	0.74 (0.23, 1.94)	0.572
Yaquis	0.9	0.33	2.76	2.47 (1.3, 4.71)	0.012
Girls*OB	−2.78	1.12	−2.49	0.06 (0.03, 0.39)	0.011
High ApoB					
Intercept	−1.59	0.4	−3.99	0.20 (0.09, 0.44)	˂0.001
Age	0.09	0.04	2.21	1.10 (1.01, 1.19)	0.032
OB	0.68	0.2	3.33	1.97 (1.32, 2.94)	˂0.001
NM	1.22	0.21	5.72	3.38 (2.24, 5.15)	˂0.001
Seris	−1.23	0.38	−3.23	0.29 (0.13, 0.59)	˂0.001
Yaquis	−0.92	0.27	−3.46	0.40 (0.23, 0.66)	˂0.001
Dyslipidemia					
Intercept	1.08	0.15	7.14	2.93 (2.20, 3.97)	˂0.001
OB	0.93	0.35	2.63	2.54 (1.31, 5.34)	0.009
NM	0.18	0.27	0.67	1.20 (0.71, 2.07)	0.500
CM-Puebla	−0.86	0.19	−4.45	0.42 (0.29, 0.62)	˂0.001
Seris	−1.41	0.3	−4.67	0.24 (0.13, 0.44)	˂0.001
Yaquis	−1.11	0.23	−4.73	0.33 (0.21, 0.52)	˂0.001
NM-Urban*OB	−1.28	0.56	−2.26	0.28 (0.09, 0.85)	0.022
CM-Puebla*OB	1.74	1.09	1.59	5.68 (0.96, 10.05)	0.112
Seris*OB	1.2	1.17	1.02	3.30 (0.45, 8.29)	0.314
Yaquis*OB	0.6	0.88	0.69	1.83 (0.37, 13.59)	0.492

Lipid disorders (high or low levels) were classified according to the borderline-high criteria of the Expert Panel on Integrated Guidelines for Cardiovascular Health and Risk Reduction in Children and Adolescents of the National Heart, Lung and Blood Institute. * interaction, na, information for ApoA1 and ApoB in CM-Puebla is not available.

## Data Availability

Raw data were generated and derived data supporting the findings of this study are available from the corresponding author PCU upon request.

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
