# Peer review of "Dyslipidemia, Obesity, and Ethnicity in Mexican Children"

_ijerph, 2021, doi:10.3390/ijerph182312659_

Round 1

Reviewer 1 Report

The present study reports the occurrence of cardiometabolic risk markers (lipid profiles) in school children from five different ethnic groups in Mexico. It also evaluates the ability of BMI and body fat percentage to predict the examined markers and analyses their association with obesity and ethnicity.

One of the constraints of the study is the lack of data on dietary habits, lifestyle, and family history of dyslipidemia, which allowed authors only to speculate on the causes of the observed differences among studied ethnic groups.

The study is interesting; still, the data presenting should be improved. Some issues that should be addressed: - The authors should carefully revise the English language of the manuscript.

In addition to several typing mistakes and grammar errors, some sentences are rather hard to follow.

  • The authors should correct the table captions and content for Tables 2 and 3; they are interchanged - Please verify the data presented in table 3 and lines 164-165; they do not fully match. - cardiometabolic risk, cardiometabolic risk markers, cardiometabolic markers, and lipid markers are used interchangeably throughout the manuscript. The authors should consider defining that cardiometabolic risk was assessed through measurements of lipid profiles in the introduction; also, the authors should consider replacing the “cardiometabolic prevalence” with a more suitable term.
  • - The authors should consider mentioning in the result section 3.2 that BFP was only measured in CM-Mexico City; it would be easier to follow.
  • - Consider using “performed better to predict” for Lines 23, 255, 301, and wherever appropriate
  • - The authors should consider improving the conclusion section and providing more information on the novelty of the data presented and their significance
  • - The abbreviations should be introduced at their first mentioning (Line 35, CVD; BMI, BFP, line 59)
  • - Line 43, consider adding the year

Author Response

We want to thank the reviewers for such constructive comments, which we addressed point-by-point along with the manuscript. We hope that the current version of the manuscript meets expectations in terms of its general improvement.

Please notice that with highlight some essential changes in blue, but the IJERPH English Editing Services edited the whole manuscript.

Reviewer 1

The present study reports the occurrence of cardiometabolic risk markers (lipid profiles) in school children from five different ethnic groups in Mexico. It also evaluates the ability of BMI and body fat percentage to predict the examined markers and analyses their association with obesity and ethnicity.

  1. One of the constraints of the study is the lack of data on dietary habits, lifestyle, and family history of dyslipidemia, which allowed authors only to speculate on the causes of the observed differences among studied ethnic groups.

Authors' response. That is right. Unfortunately, did was not possible to collect unbiased dietary patterns of the children. Almost all the work was done in schools. At first, we tried to collect data on their diet, but the children answered what they thought we wanted to hear. For this reason, we did not use this information. This limitation is mentioned in the "limitations of the study."

  1. The study is interesting; still, the data presenting should be improved. 

Authors' response. We made several modifications, and we rewrote several sentences in the text to improve the presentation. We hope the quality of the text was improved.

  1. Some issues that should be addressed: - The authors should carefully revise the English language of the manuscript. In addition to several typing mistakes and grammar errors, some sentences are rather hard to follow.

Authors' response. Sorry for that. The IJERPH English Editing Services edited the manuscript.

  1. The authors should correct the table captions and content for Tables 2 and 3; they are interchanged. Please verify the data presented in table 3 and lines 164-165; they do not fully match. – 

Authors' response. Sorry for that. It was corrected 

  1. Cardiometabolic risk, cardiometabolic risk markers, cardiometabolic markers, and lipid markers are used interchangeably throughout the manuscript. The authors should consider defining that cardiometabolic risk was assessed through measurements of lipid profiles in the introduction; also, the authors should consider replacing the "cardiometabolic prevalence" with a more suitable term.

Authors' response. We now focused the manuscript on lipids disorders instead of cardiometabolic risk to avoid the lack of concept and terminology heterogeneity. We feel the current version reflects the content and data analysis of the manuscript better. Thus, we made several terminology replacements and changed the title.

  1. The authors should consider mentioning in the result section 3.2 that BFP was only measured in CM-Mexico City; it would be easier to follow.

Authors' response. Following your suggestion, it was highlighted in the 3.2 section 

  1. Consider using "performed better to predict" for Lines 23, 255, 301, and wherever appropriate

Authors' response. Thank you for this suggestion. The replacement was made in all appropriate situations. 

  1. The authors should consider improving the conclusion section and providing more information on the novelty of the data presented and their significance.

Authors' response. Following your suggestion, we rewrite the whole conclusion section 

  1. The abbreviations should be introduced at their first mentioning (Line 35, CVD; BMI, BFP, line 59)

Authors' response. Thank you for noticing that. It was corrected.

  1. Line 43, consider adding the year

Authors' response. The period of the study was added.

Reviewer 2 Report

The findings of obesity and dyslipidaemia in five different groups of Mexican children show high prevalence of both and will be of interest to policy makers and health professionals.

The data is well collected but the presentation is excessively complicated and difficult to read so the message gets lost in the detail.

The results section starting 66% of the overall children..... is hard to correlate with the tables and the legend for table 3 is hard to correlate with the adjacent table.

Suggest the text highlight the key points rather than a lot of detail and refer to the relevant tables allowing interested readers to look at the tables themselves for more details.

Cardiometabolic risk normally includes BP and may include Glucose both of which could be increased in this population. 

The title could be changed to "Obesity, dyslipidaemia and ethnicity in Mexican children" to better reflect this and the text including abstract, methods, results and discussion could also be changed to reflect this. This does not invalidate a good study but its an important limitation.

BFP should be defined when it first appears including relevant cut off values

Author Response

We want to thank the reviewers for such constructive comments, which we addressed point-by-point along with the manuscript. We hope that the current version of the manuscript meets expectations in terms of its general improvement.

Please notice that with highlight some essential changes in blue, but the IJERPH English Editing Services edited the whole manuscript.

The findings of obesity and dyslipidemia in five different groups of Mexican children show high prevalence of both and will be of interest to policymakers and health professionals.

  1. The data is well collected, but the presentation is excessively complicated and difficult to read, so the message gets lost in the detail.

Authors' response. Sorry for that. The IJERPH English Editing Services edited the manuscript. We also added a figure to give more clarity of the results

  1. The results section starting 66% of the overall children is hard to correlate with the tables, and the legend for table 3 is hard to correlate with the adjacent table. Suggest the text highlight the key points rather than a lot of detail and refer to the relevant tables allowing interested readers to look at the tables themselves for more details.

Authors' response. We rephrase several sentences of the result section the make the reding more fluid 

  1. Cardiometabolic risk normally includes BP and may include Glucose, both of which could be increased in this population. 

Authors' response. We now focused the manuscript on lipids disorders instead of cardiometabolic risk to avoid the lack of concept and terminology heterogeneity. We feel the current version reflects the content and data analysis of the manuscript better. Thus, we made several terminology replacements and changed the title.

  1. The title could be changed to "Obesity, dyslipidemia and ethnicity in Mexican children" to better reflect this and the text including abstract, methods, results and discussion could also be changed to reflect this. This does not invalidate a good study but its an important limitation.

Authors' response. Thank you for your suggestion. Following your suggestion, we changed the title and focused on the dyslipidemias concept instead of the cardiometabolic concept.

  1. BFP should be defined when it first appears including relevant cut off values

Authors' response. Thank you for noticing that. It was corrected.

Round 2

Reviewer 1 Report

The authors answered all the raised issues satisfactorily.

Author Response

Thank you so much

Reviewer 2 Report

Thank you for the revised version which has adressed my concerns and is now much easier to read.

My only request is that you clarify the brick like multicoloured panels above (adjacent to A, B, C etc)  and below the blue and pink bars

Author Response

Reviewer

My only request is that you clarify the brick like multicoloured panels above (adjacent to A, B, C etc)  and below the blue and pink bars

Authors: Thank you so much for the comment. We clarified the colors in the legend and in the text
